# Abrupt transitions in Arctic open water area

Michael A. Goldstein<sup>1,2</sup>, Amanda H. Lynch<sup>3,4</sup>, Todd E. Arbetter<sup>3</sup> and Florence Fetterer<sup>5</sup>

<sup>1</sup>Climate Change Research Center, University of New South Wales, Sydney NSW 2052 Australia.

<sup>2</sup>Finance Division, Babson College, Babson Park, MA 02457 USA.

<sup>3</sup>Institute at Brown for Environment and Society, Brown University, Providence, RI 02912 USA.

<sup>4</sup>Department of Earth, Environmental and Planetary Sciences, Brown University, Providence, RI 02912 USA.

<sup>5</sup>National Snow and Ice Data Center, Cooperative Institute for Research in the Environmental Sciences, University of Colorado, Boulder, CO 80309 USA.

10

5

Correspondence to: Amanda H. Lynch (<u>Amanda Lynch@brown.edu</u>)

**Abstract.** September open water fraction in the Arctic is analyzed using the satellite era record of ice concentration (1979-2014). This analysis suggests that there is a statistically significant breakpoint (shift in the mean) and increase in the variance

- around 1988 and another breakpoint around 2007 in the Pacific sector. These structural breaks are robust to the choice of algorithm used for deriving sea ice concentration from satellite data, and are also apparent in other measures of open water, such as operational ice charts and the record of navigable days from Barrow to Prudhoe Bay. Breakpoints in the Atlantic sector record of open water are evident in 1988 and 2007 but more weakly significant. The breakpoints appear to be associated with concomitant shifts in average ice age, and tend to lead change in Arctic circulation regimes. These results
- support the thesis that Arctic sea ice may have critical points beyond which a return to the previous state is less likely.

# **1** Introduction

In the most recent decade, summer minimum sea ice extent has retreated to levels not seen since the beginning of the satellite record [*Fetterer et al.*, 2002]. With their geographic proximity to ice edge areas, the development of Arctic coastal communities – from *Inuit Nunangat* (homeland) to whaling station to national security frontline – continues to be contested.

- Low global oil prices, along with technological and security challenges connected to production, may render Arctic fossil fuel reserves economically nonviable for the foreseeable future [*Mohn and Osmundsen*, 2008; *Krauss and Reed* 2015]. But even without significant natural resource development, industries such as freight, tourism and commercial fishing may benefit from lower ice area, thickness, and concentration. These activities have a critical reliance on shipping in the Arctic, but because of the present low predictability of the navigable season, economic benefits have not accrued at the expected rate
- [Krupnik and Jolly, 2002; AMAP, 2011]. Low predictability has also led to ongoing safety and reliability concerns [Liu and Kronback, 2010]. Indeed, even though summer sea ice is projected to have substantially retreated by as early as 2035, operators will face continued hazards from drift ice, icebergs and potentially increased storminess [Parkinson and Comiso, 2013].

The confluence of opportunity and risk at the retreating ice edge [*Lei et al.*, 2013] raises critical questions as to how well we observe and simulate Arctic open water area. That is, how well do we describe the area covered by ocean within which ice, if present at all, is present in very low concentrations? In this context, the reconstruction of historical ice area records has been an important activity [e.g. *Rogers et al.* 2013]. However, much of the focus has been upon hemispheric ice extent and

- 5 concentration, whereas more critical to Arctic industrialization is the area and timing of seasonally navigable open water. While this quantity is of course dependent on ship type, crew experience, and weather conditions, a decadal-scale record of open water provides a first order understanding of how sea ice will affect Arctic economic development over coming decades.
- In this paper, we examine the record of open water area with a focus on the period 1979 to 2014 across the Arctic Ocean and its peripheral seas. We compare the records derived from satellite data using different analysis algorithms, and compare these records to other measures of open water. We demonstrate significant spread among the records, but all reveal key structural shifts both in the mean and in the variability. These shifts are robust to satellite analysis algorithm but are limited geographically, and can be linked to sea ice age and atmospheric variability. The results suggest that breakpoints in ice regimes have already occurred in the record, although artifacts arising from the brevity of temporally consistent records
- regimes have already occurred in the record, although artifacts arising from the brevity of temporally consistent records cannot conclusively be ruled out. This finding suggests caution must be employed when seeking to understand trends over the satellite era and to reproduce these trends in operational and climate models. Further, it calls into question the practice of characterizing the record using linear trends.

# 2 Data

- The primary data source is monthly sea ice concentration from the combined Nimbus Scanning Multichannel Microwave Radiometer (SMMR: 1979–1987), the Defense Meteorological Satellite Program (DMSP) Special Sensor Microwave/Imager (SSM/I: 1987–2007) and the Special Sensor Microwave Imager/Sounder (SSMIS: 2008-present) 25-km gridded sea ice concentration data products. Detailed analysis is conducted with four sea ice concentration datasets: NASA Team algorithm data [*Cavalieri et al.*, 1996]; Bootstrap algorithm data [*Comiso*, 2000]; a Climate Data Record covering 1988-2014 [*Meier et Comiso*, 2000]; a Climate Data Record covering 1988-2014 [*Meier et Comiso*, 2000]; a Climate Data Record covering 1988-2014 [*Meier et Comiso*, 2000]; a Climate Data Record covering 1988-2014 [*Meier et Comiso*, 2000]; a Climate Data Record covering 1988-2014 [*Meier et Comiso*, 2000]; a Climate Data Record covering 1988-2014 [*Meier et Comiso*, 2000]; a Climate Data Record covering 1988-2014 [*Meier et Comiso*, 2000]; a Climate Data Record covering 1988-2014 [*Meier et Comiso*, 2000]; a Climate Data Record covering 1988-2014 [*Meier et Comiso*, 2000]; a Climate Data Record covering 1988-2014 [*Meier et Comiso*, 2000]; a Climate Data Record covering 1988-2014 [*Meier et Comiso*, 2000]; a Climate Data Record covering 1988-2014 [*Meier et Comiso*, 2000]; a Climate Data Record covering 1988-2014 [*Meier et Comiso*, 2000]; a Climate Data Record covering 1988-2014 [*Meier et Comiso*, 2000]; a Climate Data Record covering 1988-2014 [*Meier et Comiso*, 2000]; a Climate Data Record covering 1988-2014 [*Meier et Comiso*, 2000]; a Climate Data Record covering 1988-2014 [*Meier et Comiso*, 2000]; a Climate Data Record covering 1988-2014 [*Meier et Comiso*, 2000]; a Climate Data Record covering 1988-2014 [*Meier et Comiso*, 2000]; a Climate Data Record covering 1988-2014 [*Meier et Comiso*, 2000]; a Climate Data Record covering 1988-2014 [*Meier et Comiso*, 2000]; a Climate Data Record covering 1988-2014 [*Meier et Comiso*, 2000]; a Clima
- *al.*, 2013]; and a separate "merged" concentration from *Meier et al.* [2013], covering 1979–2014.

The Climate Data Record (CDR) meets strict requirements that it be reproducible. The "merged" concentration record is produced in the same manner as is the CDR, but for the period prior to 1988, it incorporates data that underwent manual quality control and for that period it is not strictly reproducible, and therefore cannot be called a CDR. The passive microwave satellite products do not detect ice well when it is at low concentrations and for this reason grid cells at less than

30 microwave satellite products do not detect ice well when it is at low concentrations and for this reason grid cells at less than 15% concentration are considered to be open water for the purposes of this study. We extend this definition of "open water"

to the data sets described below, even though they may record ice at low concentrations more accurately than the satellite products.

Also analyzed is the Hadley Centre Sea Ice and Sea Surface Temperature dataset (HadISST2.2) [Titchner and Rayner, 2014]. For the first half of the 20th century this dataset relies largely on the Arctic and Southern Ocean Sea Ice 5 Concentrations [Chapman and Walsh, 1991] which in turn makes use of sea ice charts from a Danish source [DMI and NSIDC, 2012] as described in Walsh and Chapman [2001]. These charts are based on direct observations from the shoreline and ships [Kelly, 1979], and hence tend to overestimate ice extent in regions inaccessible by sea. Further, these charts lack primary data for the time period 1940-1952. Hence, the dataset is used only for the time period 1953–2014, when sources expanded to include aerial surveys, observational re-analyses, operational ice charts, and from 1979, the NASA Team

10

product.

The National Ice Center (Washington, DC) produces operational ice charts for the Arctic on weekly to biweekly intervals. An individual chart, covering a sub-region of the Arctic, such as the Beaufort Sea or Laptev Sea, is prepared by an ice analyst using all available forms of remotely sensed sea ice data, including the passive microwave data that is used to 15 produce the automated NASA Team and Bootstrap near real-time products, visible and near-infrared products such as MODIS, and synthetic aperture radar such as RADARSAT-2. A chart for a given week may consider any data acquired that week and contains more information, such as partial concentrations by ice type, than a satellite retrieval. Here, we use the ice extent (defined as the area covered by ice at greater than 15% concentration) from the 1972–2007 archived product [National Ice Center, 2006; 2009] for the last chart produced in September of each year. In processing the ice chart data for this study, 20

any grid cell with ice concentration less than 15% was set to open water to be consistent with the satellite retrievals.

For each satellite data record and each year in the record, a mean September open water area within a Pacific and an Atlantic sector is determined. The Pacific sector is bounded by 100°W, 100°E, 70°N, and 90°N. For the Atlantic sector boundaries are 100°E, 100°W, 80 °N and 90°N. Within each sector and for each day, the number of grid cells for which the ice 25 concentration is less than 15% is tallied and the area covered by those cells is computed. A monthly average is then computed from these daily open water areas. HadISST data and National Ice Center sea ice chart data were processed in the same way, although the National Ice Center charts were not available as daily data.

Additional data were obtained for analysis. These include the ERA-Interim daily mean sea level pressure (SLP) anomaly 30 data for 1979-2014, interpolated to the 50 km EASE (Equal-Area Scalable Earth) grid [Dee et al., 2011]. Also examined was the record of navigable days for the Barrow-Prudhoe Bay sea route obtained from the National Ice Center [Barnett, 1976, 1980] - this record covers 1953–2013 and assumes conditions are navigable when sea ice concentrations are less than or equal to 50%. The data is described in some detail in Drobot and Maslanik [2003].

5

25

The EASE-Grid Sea Ice Age Version 3 [*Tschudi et al.*, 2016] is a sea ice age product on a 12.5 km subset of the EASE grid. It is based on retrievals from several satellites (AMSR-E, AVHRR, SMMR, SSM/I, SSMIS) as well as drift buoy data (IABP), and is augmented by NCEP/NCAR re-analyses. Using a Lagrangian tracking method, individual parcels of ice are identified and tracked in weekly time steps as they move through the Arctic. These are rasterized onto a 12.5 km grid with the oldest ice in the cell assigned as the value for that cell, following the assumption that younger, thinner ice is

preferentially removed by convergence (ridging) or divergence. The age of the ice is an integer value by year that is incremented following the end of the melt season if at least 15% of the ice survives. While the tracking method begins with the start of SMMR data in November 1978, ice ages are only provided from the first week of 1984.

## **3** Methods

- Several statistical tests were used to determine whether the data records are characterized by a trend as described by, e.g. Danielson et al., [2011] or a shift in the mean (that is, a structural change or breakpoint). Structural changes in the ice cover record have been suggested previously. In the eastern Bering Sea, a late 70's regime shift in the long term sea ice cover record developed by *Walsh and Johnson* [1979] was associated by *Niebauer* [1998] with a 50 to 70 year oscillation in the North Pacific. In future climate scenarios, *Holland et al.* [2006] detect abrupt changes in September ice extent and
- thickness that are characterized by thermodynamically driven doubling of ice loss for periods of around five years, in seven of sixteen models participating in the A1B scenario of the IPCC-AR4. In our analysis, the data records are split into sub-periods. The mean for each of the sub-periods was calculated and these means were tested against the linear regression for the entire record as explanatory variables. Following an exploratory analysis, detailed investigation, as described in the results section, focuses on three sub-periods (1979-1988, 1989-2006 and 2007-2014) in the NASA Team record and record
- of navigable days for the Barrow-Prudhoe Bay sea route obtained from the National Ice Center.

To reduce the dimensionality of the sea level pressure records, a self-organizing map analysis was conducted, consistent with the approaches in *Mills and Walsh* [2014] and *Lynch et al.* [2016]. July, August and September daily sea level pressure anomalies (that is, summer monthly differences from the entire period summer mean) for the region north of the Arctic circle and below 500 m elevation were calculated and allocated through an unsupervised neural network approach to a 5x4 array of nodes (Figure 1). The frequency of days allocated to each node was found for each year in the record. In addition, monthly mean open water anomalies for September from the NASA Team record were binned into classes ranging from greater than 1.5 standard deviations (std), between 0.5 and 1.5 std, between 0.5 and -0.5 std, between -0.5 and -1.5 std and less than -1.5

- std, The occurrence of these classes was allocated to each node, to determine the Arctic circulation regimes associated with
- open water extremes, following the approach of Lynch et al. [2016].

## 4 Trends and Variability in Records of Open Water

The open water fraction area is calculated for the Atlantic and Pacific sectors for each of the satellite derived products, as well as the HadISST product and the operational ice charts (Figure 2). The records shows substantial differences that arise from the choice of algorithm to process satellite microwave brightness temperatures to ice concentration, and from the use of additional source data such as ship and aircraft observations in the case of HADISST and NIC ice charts. *Bunzel et al.* [2016] found that these differences are large enough to affect seasonal forecast skill, causing disparities of the order of 2°C in a five month forecast when used for initialization. The characteristics of passive microwave remote sensing that lead to these differences, while well documented [e.g. *Comiso et al.*, 1997; *Ivanova et al.*, 2014], are particularly sensitive when the measure is September open water [*Hwang and Barber*, 2006]. However, this quantity is highly decision relevant, and hence assessments of predicted open water season using the satellite record itself are critical, particularly for physical sea ice models that use passive microwave data for initialization or verification.

The NASA Team algorithm is known to perform best with high-concentration multiyear, wintertime ice when compared to the Bootstrap algorithm [*Comiso et al.*, 1997; *Meier*, 2005]. The Bootstrap algorithm is considered to be more reliable in cases of surface melt or when concentrations are lower than about 40%, both of which are likely to be occurring in September. Therefore, it is not surprising that the Bootstrap algorithm results in systematically lower open water area than the NASA Team algorithm. The CDR and merged products are obtained by comparing the ice concentrations derived by the NASA Team and Bootstrap algorithms and choosing the higher concentration of the two, reasoning that either method may underestimate the ice area at some location at some time of the year. For our study, the CDR generally follows the Bootstrap

- 20 curve, as does the merged product. Slight differences in the numbers for Bootstrap (GSFC) and NASATeam (GSFC) are attributable to slightly different radiances used in the calculations done at NSIDC versus those done at NASA (for more details, see *Meier et al.* [2013]). The HadISST record shows systematically lower open water than any of the satellite-derived records. This is due to two primary factors. The first is that the HadISST product incorporates the NIC charts, which in the Atlantic sector during high ice years generally report more ice cover than the satellite-derived records. The second is
- that the HadISST product uses 1 degree grid spacing rather than the 25 km used in the satellite-derived records. This lower resolution results in few grid cells reporting below 15% ice concentration over the entire cell.

An exploratory analysis of these time series found that while magnitude varies significantly across the records, the NASA Team (NSIDC) open water record is representative in its trends and variability, and hence we focus on this single time series

30 for clarity. It is apparent that there is a weak linear trend (2% per decade, adjusted  $R^2=0.40$ ) in the Atlantic sector but a large and significant linear trend (9% per decade, adjusted  $R^2=0.64$ ) in the Pacific sector (Figure 3, trend lines not shown). However, a linear regression may not be the best model for this data record. There is an ostensible breakpoint at 1988 in the September open water record on the Pacific side that is not apparent on the Atlantic side (nor when the whole Arctic is

5

analyzed, not shown). Further, the inter-annual variability in the Pacific sector is much larger after 1988 (around 11.4%) than it is before 1988 (around 2.0%), while the Atlantic shows a decline in variability (3.7% before; 3.2% after). A second breakpoint can be identified in 2007 in the Pacific sector that is not evident in the Atlantic sector. In order to determine the most robust model for the Pacific sector record, the means of the Pacific sector from 1979-1988, 1989-2006, and 2007-2014 were calculated. A model comprised of a constant, a trend variable, and a variable with the three means for each period had an adjusted  $R^2$  of 0.735. The coefficients for both the constant and the trend were insignificant (p-values of 0.276), while the variable with the three means is highly significant (p-value of 0.0009). A regression with just a constant and the variable with the three means has an adjusted  $R^2$  of 0.734, which is higher than the adjusted  $R^2$  of 0.639 for a regression with a

constant and a trend variable. This analysis suggests that the best model for the record of open water in the Pacific sector 10 may indeed be a series of abrupt shifts rather than a linear regression.

The short record of satellite-derived time series prior to 1988 is a limitation of this model. The NIC chart-derived time series from 1972-2007 was tested for the Pacific and the Atlantic sectors. Because this record stops in 2007, only one breakpoint, in 1988, could be assessed. It was found that both models – a single trend variable or one mean break (1972 to 1988, 1989 to 2007) – were insignificant in combination, and equally valid when tested separately. The adjusted  $R^2$  for the Pacific was

15 2007) – were insignificant in combination, and equally valid when tested separately. The adjusted  $R^2$  for the Pacific was around 0.29 and for the Atlantic was around 0.25. That is, either a linear trend or a structural break in 1988 was a similarly legitimate model of the NIC chart time series. No other breakpoints were identified in this record in either sector.

The HadISST time series and the number of navigable days on the Barrow-Prudhoe Bay route both present a longer record, although both also have shortcomings. The HadISST record is not temporally uniform, and has a lower spatial resolution than the satellite derived records. In the Pacific sector, the HadISST time series reflects the NASA Team results. A model comprised of a constant, a trend variable, and a variable with the three means for each period had an adjusted R<sup>2</sup> of 0.74. The coefficients for both the constant and the trend were insignificant (p-values of 0.49), while the coefficient for the three means is highly significant (p-value of 0.0001). More interestingly, the results for the Atlantic sector, while weak, at just over the

- 25 5% significance level are technically the same as the Pacific sector. Here, the model that combines a constant, trend, and a variable with the three means has an adjusted  $R^2$  of about 0.48. In this model, the coefficients for both the constant and the trend were almost significant: the constant had a p-value of 0.060 and the trend has a p-value of about 0.057. This is not significant at the 5% level, but is significant at the 10% level. In this model, the coefficient on the three means has a p-value of 0.0049, or significant at the 1% level. The adjusted  $R^2$  for the Atlantic model with just a constant and a trend is
- 30 0.41; the adjusted R<sup>2</sup> for the Atlantic model with just a constant and the three means is higher at 0.45, again supporting a better fit by a model with two structural breaks rather than a linear trend. That is, the model of two breakpoints in the Pacific sector suggested in the satellite derived data is verified in the HadISST data. However, this model could also be argued to be marginally true also for the Atlantic sector, and at the very least, there is evidence of both breakpoints and trends.

The Barrow-Prudhoe Bay navigability time series is temporally uniform. Importantly, as noted, open water in itself is not an accurate measure of navigability, which depends on many factors. Most critically, the type of vessel to be employed has impact on navigability criteria. Vessels may be rated with an ice class under the International Association of Classification

- Societies Unified Requirements for Polar Class Ships, but this is not mandatory. The international standard uses ice age, as a proxy for thickness, and season, rather than ice concentration, to differentiate between vessels [Rogers et al. 2013]. Other classifications include the Finnish-Swedish ice class used in the Baltic Sea and the American Bureau of Shipping ice classes. The Barrow-Prudhoe Bay route is considered navigable for ice up to 50% in concentration in good weather, due to the small volume of traffic and the proximity of the coastline. The number of days in the season that the entire Barrow-Prudhoe Bay
- route is navigable is shown in Figure 4, along with the model that assumes a shift in the mean from 1988 to 1989 and again from 2007 to 2008. The model comprised of a constant, a trend variable, and a variable with the three means for each period had an adjusted  $R^2$  of 0.189. The values for the constant and the coefficient on the linear trend (both had p-value of 0.497) were not significant, but the coefficient on the variable with the three means was significant at the 5% level (p-value of 0.0145). While both the intercept and the positive trend coefficients were significant for the trend regression for the entire
- period, when trend regressions were run over the three sub-periods, (1953-1988, 1989-2007 and 2008-2013), the first two sub-periods had a negative adjusted  $R^2$  (and the final period had an adjusted  $R^2$  of only 0.067) and in each sub-period regression both the intercept and the trend coefficients were insignificantly different from zero (the coefficients on the trend were negative in sign, but not significant), suggesting that there was no trend during these sub-periods. This analysis provides further evidence that a model of the open water time series that includes breakpoints in 1988 and 2007 is more
- significant than a linear trend in the Pacific sector.

### **5** Discussion

The model of abrupt shifts in September open water fraction is likely to have a number of drivers that interact in complex ways. First, it is possible that non-physical artifacts may exist in the satellite record. *Bjorgo et al.* [1997], *Comiso and Nishio* [2008] and *Cavalieri et al.* [2012] note small artifacts remain in the record associated with the sensor and orbit changeover

- between SMMR and SSM/I, but all suggest that these differences are below the sensitivity of the instrument. That said, it is known that there were some errors in the data stream in the first months of the SSM/I instrument (J. Stroeve, pers. comm.; W. Meier, quoted in *Harvey* [2016]). While this may have some impact on the structural shift detectable in 1988, given that a breakpoint was also detected in 2007 there are likely to be other, physical, factors that are important.
- The hypothesis that the ice volume reaches a critical threshold allowing a critical point and subsequent rapid melting is one that has been identified in model studies [e.g. *Holland et al.* 2006]. Ice age is a useful, though not perfect, proxy for ice thickness. Figure 5 shows anomalies of ice age for Septembers (for week 37), where anomaly is defined relative to a base

period of 1984 through 2014. Note that ice age anomalies increase rapidly in the first 6 years of the record, although this is likely an artifact of the data product's initialization. After 1988, multiyear ice with concomitant positive anomalies in both the Pacific and the Atlantic dwindles, particularly after the successive record low summers of 2007 and 2012. Very little multivear ice remains; in fact, since the algorithm favors the oldest ice in the grid cell, the amount of old ice in recent data

- may be overestimated. Nevertheless, there is a strong relationship between the ice age record and the open water area (also 5 shown in Figure 5). A linear regression shows that these records are correlated at 0.56 in the Pacific sector and 0.44 in the Atlantic sector. A model for ice age in the Pacific sector with a constant, a linear trend, and a variable with three means separated by breakpoints in 1988 and 2007 has an adjusted  $R^2=0.695$ . The coefficients on the constant and the linear trend variables were not significant at the 5% level (both p-values around 0.083), but the coefficient on the variable for the three
- 10 means was significant at more than the 1% level (p-value 0.0016) Similarly, in the Atlantic sector, a model with a constant, a trend, and the three means variable had an adjusted  $R^2=0.539$ , and the coefficient on the three means variable was significant at the 5% level (p-value=0.02) while the trend is not (p-value 0.088). Because of the initialization of the ice age record, it is not possible to ascribe a structural shift unambiguously in 1988. However, it appears that the abrupt increase in open water area is associated with a concomitant abrupt decrease in ice age across the Arctic. Further, it is possible that 2007 represents something of a critical point for the sea ice, particularly in the Pacific. 15

An abrupt shift in Arctic sea ice also raises the question as to the role of atmospheric circulation. We focus on the Pacific sector, where the appropriate statistical models are most unambiguous. Further, small shifts in circulation can have a particularly large impact on ice cover in the Pacific sector, where a slight shift in wind direction can make the difference between coastal accumulation on the one hand and export and melt on the other [e.g. Maslanik et al., 2000]. In an analysis of 20 the most extreme Pacific sector open water years in the record, Node (3,1) in Figure 1 is most typical of the associated circulation. Consistent with the analysis by Lynch et al. [2016], this circulation regime is characterized by warm southerlies originating over Canada and Alaska, warm temperatures and low ice area over the Beaufort Sea, and low terrestrial snow extent in the Canadian Arctic. The analysis of Lynch et al. [2016] also demonstrated that this relationship between open water and atmospheric circulation is consistent over time. Ice advection in this regime is characterized by a strong transpolar 25 transport and export through Fram Strait. Figure 6 shows the frequency of occurrence of Nodes (3,1), (4,1), (3,2) and (4,2). Note that nodes situated in the same region of the self organizing map are broadly similar, and all of these nodes are associated with varying degrees of positive open water anomalies. Figure 6 shows that these circulation regimes – associated with a warm Pacific - do not occur at all in 1990 and 2011. Each of these years are followed by circulation regimes that

favor the more intense open water anomalies - that is regimes that favor increased inter-annual variability. These 30 relationships suggest that the open water anomalies lead the atmospheric circulation anomalies.

A long-term, high quality and temporally consistent record of Arctic open water remains an elusive goal. That said, there is much insight that can be derived from a careful analysis of all of the available data. This assessment of satellite-derived,

operational, and climatological time series suggests that Arctic sea ice behavior is highly non-linear, and critical points in ice cover that have been postulated as arising from strong ice-albedo feedback processes may have already occurred. We have demonstrated that, certainly in the Pacific sector and perhaps for the Atlantic sector, the fraction of open water in the marginal ice zone is better characterized by a series of structural shifts, in 2007 and in 1988, rather than a linear trend. This

- model is robust to the algorithm used to derive ice concentration from satellite passive microwave brightness temperatures, although these algorithms demonstrate significant differences in magnitude. The physical nature of these structural shifts is supported by analysis of quasi-independent records, such as the operational ice charts, and the navigability of the Barrow-Prudhoe Bay sea route. The concomitant shifts in the ice age product suggests a potential thermodynamic feedback mechanism. That said, the processes associated with these shifts can only be tested independently using a modeling
- approach, which will be the subject of future work.

### Acknowledgments and Data

This work was partially supported by a Visiting Fellowship from the Cooperative Institute for Research in Environmental Sciences. All data used in the analysis are freely available from the National Snow and Ice Data Center with the exception of the HadISST2 dataset, which was obtained from the UK Met Office, and the Barrow-Prudhoe Navigability Data, which was

15 provided by Christopher Szorc of the US National Ice Center. The authors acknowledge with thanks the technical support of Alexander Crawford and comments from Julienne Stroeve. The authors identify no financial conflicts of interest involved in this work.

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
