# Peer review of "Abrupt transitions in Arctic open water area"

_The Cryosphere, 2016_

## Referee Comment (RC1) · Anonymous Referee #1 · 21 Jun 2016

Summary

This paper presents an analysis of sea ice extent derived from satellite passive microwave data and other sources to investigate changes in open water area in the Arctic. Transitions found in 1988 and 2007, particularly in the Pacific sector. The results suggest that there may be critical thresholds in the Arctic system where a return to a previous state is unlikely.

General comment

Overall, this is an interesting paper and addresses an important issue. The analysis is solid and the paper is well written. I think my main concerns are with the source data. For example, overall, the quality of the passive microwave data is good and there has been considerable effort for the record to be consistent over time. However, some

inconsistencies may remain, particularly regionally. Eisenman et al., The Cryosphere, (2014), for example found potential inconsistencies; these were primarily in the Antarctic, but in supplemental material, potential Arctic inconsistencies are also found.

One thing that is a particular concern is the periods used for the "break" points. How were these chosen? Were they ad hoc, or derived from the behavior of the data. Based on the text, they seem ad hoc. If so, that seems arbitrary. If not, then this should be made clearer in the text when the periods are first discussed (pg. 4, line 19). Also, I wonder about the periods being of different length – 9 years, 17 years, and then 7 years – this also has an ad hoc feel to it.

Getting back to the satellite transitions, these periods are a particular concern because they seem to at least roughly coincide with satellite transitions, particularly the 1988 transition, which roughly corresponds with the switch in sensors from SMMR to the SSMI series. SMMR and SSMI are different sensors, with different channels, so there are some distinct differences in the sensors that may affect sea ice concentrations – e.g., sensor resolution (sensor footprint), frequencies, etc. The algorithms are adjusted to try to account for these differences, but it's not possible to do so exactly. This seems particularly apparent in Figure 3, where the SMMR record for the Pacific section show much lower variability than the SSMI and SSMIS era. I would guess that this is likely an artifact of the different sensors. There was also a sensor transition in 2008, between SSMI and SSMIS, though the differences between the two are much smaller.

The authors do a good job in not simply relying on the passive microwave record, but also looking at the Hadley sea ice and the NIC ice charts. However, both Hadley and the NIC charts rely on passive microwave data as source. Hadley uses passive microwave for the period after 1978. And NIC charts were reliant on passive microwave for a lot of their information from 1978 until the launch of Radarsat in 1995.

I suspect that these issues do not invalidate the results but they may well have some effect. I think these issues should be addressed more in the manuscript.

[Figure]

I think this manuscript is acceptable for publication after minor revisions to address the comments here.

Other comments (by page and line number):

2, 24-25: Note that the CDR simply combines the NASA Team and Bootstrap estimates, so the different products are not completely independent.

2, 27: The "merged" product uses data that underwent manual quality control over its entire record, 1978-2014, not just the period prior to 1988.

2, 30-31: It is true that PM products do not detect ice well at low concentrations, but the 15% concentration threshold is also selected based on the relatively low spatial resolution of the sensors, which makes any ice edge somewhat "fuzzy". I believe the 15% threshold was originally defined because it was found to best match the "true" ice edge during validation studies.

3, 10: I think HadISST2 uses OSI-SAF concentrations since 1979, at least according to Titchner et al., 2014.

3, 13: It's worth noting that Radarsat data started in 1995. This represents a substantial change in NIC's chart. Before then, they used a lot of PM data; after, it was much less. Also it would be good to include a journal reference for the NIC charts, e.g., Dedrick et al., Can. J. Rem. Sens., 2001, doi:10.1080/07038992.2001.10854887.

4, 4: Not sure what you mean by "rasterized"? The source data are already on a grid. Maybe rephrase this sentence.

5, 23: The HadISST does use NIC charts, but only as a bias correction.

5, 25: I think the resolution likely accounts for a lot of the differences between the 100 km HadISST and the 25 km PM records.

7, 15: It would seem more relevant to use the same period as the PM record, i.e., 1979-1988 instead of 1953-1988, etc. Would that potentially affect the analysis?

7, 25: Adding the Eisenman reference (above) somewhere here would be useful

7, 26: What is meant by "errors in the data stream in the first months of the SSMI instrument"? Do you mean the first SSMI, launched in 1987, on DMSP F8? I'm not aware of such errors. The Harvey reference mentions recent errors in 2016, but these don't seem relevant to analysis at hand.

7, 33: The authors say Week 37 is used for ice age. Is this consistent past the sea ice "birthday" – i.e., the week when the ice is aged another year older. The week of this transition varies, depending on when the minimum occurs (I think it generally varies between week 35 and 38). It is important to be consistent throughout the record – either all years before the "birthday" or all after. If you just pick a week in September, you may capture a mixture of years before and after the birthday, which will lead to inconsistencies.

8, 4: The age algorithm doesn't so much "favor" the oldest ice as it effectively records only the oldest ice in a given cell.

Figure 2: The caption says the units are % open water, but the y-axis is 0-0.8. I assume the y-axis is actually sea ice fraction (unit-less, 0-1 range)?

Figure 3: Same comment as for Figure 2. Both figures would also benefit from a bigger font for the axis numbers and legend.

References: Just a style comment and this may be a formatting standards issue, but it would be helpful to indent the Reference list. It's very hard to pick out individual references.

---

## Author Comment (AC1) · 8 Jul 2016

A full response to major and minor comments by the reviewer is attached in a PDF. What follows is a response to the major comments by Reviewer 1.

We thank the reviewer for the useful comments and recommendations, which have helped clarify the manuscript.

We had missed the supplemental material in Eisenman et al. (2014) that examined the inconsistencies between different versions of the Bootstrap and NASA Team algorithms for the Arctic. This is an important omission, and we have included references to the paper and the supplemental material in the revised manuscript. Of the fields considered in the supplement, ice extent findings are analogous to the open water area measure we are using. We note that we use the 95% and 99% confidence intervals, which are well within the 68% error bars suggested by, for example Comiso and Nishio

(2008) and 90% confidence interval used by the IPCC AR5.

Comiso, J. C. and Nishio, F. 2008. Trends in the sea ice cover using enhanced and compatible AMSR-E, SSM/I, and SMMR data. Journal of Geophysical Research, 113, C02S07, doi:10.1029/2007JC004257. Eisenman, I., W.N. Meier and J.R. Norris. 2014a. A spurious jump in the satellite record: has Antarctic sea ice expansion been overestimated? The Cryosphere, 8, 1289-1296, doi:10.5194/tc-8-1289-2014. Eisenman, I., W.N. Meier and J.R. Norris. 2014b. Supplement of: A spurious jump in the satellite record: has Antarctic sea ice expansion been overestimated? The Cryosphere, 8, 1289-1296, doi:10.5194/tc-8-1289-2014-supplement.

We apologize for the brevity of the description of the exploratory phase of the analysis. We elaborate on the process here and have added this to the text. There are several different ways to test for structural changes in a time series. We used several and checked using an additional one on the basis of this review. As noted in the text, in an exploratory mode, we split the record into sub-periods and tested the differences in the means. In addition, we examined the data by looking for the years that displayed the largest change in one moving three year average to the next. The two largest changes that were already not included in a previously identified six year period (two three-year moving average periods) using this measure were 1988-1989 and 2006-2007. The next stand-alone change had a much smaller change and the magnitude of the change was relatively similar to the next few candidates. While there is no particular justification for choosing the top 2, 3 or N values, the first two stood out as noticeably different in magnitude than the ones following and suggested reasonably long periods. We also ran a statistical test of the quality of various models of the time series. For example, we tested two structural shifts using a model that compared the statistical significance of a model that defined three different means and no trend to a model that defined a single trend. The former model was significant and the trend was not.

In order to increase confidence in the signal, we have now applied the method of Rodionov et al. (2004) to the NASA Team (NSDIC) record, which yielded the same two

breakpoints. While it also suggested a third breakpoint around 2001 (which is also suggested in the graph in Figure 3), using that breakpoint would create a rather small structural break. In addition, a breakpoint at around 2001 was not as strongly supported by the three year rolling average test. We therefore limited our analysis to two breakpoints. We have also added this new analysis to the paper.

Rodionov, S.N. 2004. A sequential algorithm for testing climate regime shifts. Geophysical Research Letters, 31, L09204, doi:10.1029/2004GL019448.

As noted in the manuscript, however, the short record of the satellite derived time series is a limitation. In addition, as raised by Eisenman et al. (2014) and the reviewer, there is potential for processing artifacts. As noted on page 7, line 31, we do not rule out the possibility that these shifts may be associated with artifacts in the satellite record. As the reviewer observes, it is difficult to obtain a truly independent record of the Arctic sea ice record, which is why we often use the term "quasi-independent". That said, the Barrow-Prudhoe Bay navigable days time series is a good candidate for independence, and yielded the same behavior. Nevertheless, we agree that it is not possible to completely rule out the conclusion that these shifts are not driven by physical processes. We feel that we have taken this as far as we can in the data realm and are conducting ongoing analysis using a model.

Finally, if as we suggest the structural shifts are physical in nature, there is no reason to expect that they should be periodic. If these shifts represent, for example, tipping points in ice thickness, then there is no inherent oscillation in the system that is likely to generate this behavior (although of course that is an hypothesis that could be tested in a climate model.)

Please also note the supplement to this comment:
http://www.the-cryosphere-discuss.net/tc-2016-108/tc-2016-108-AC1-supplement.pdf

**Supplement:**

Comiso, J. C. and Nishio, F. 2008. Trends in the sea ice cover using enhanced and compatible AMSR-E, SSM/I, and SMMR data. *Journal of Geophysical Research*, 113, C02S07, doi:10.1029/2007JC004257.
Eisenman, I., W.N. Meier and J.R. Norris. 2014a. A spurious jump in the satellite record: has Antarctic sea ice expansion been overestimated? *The Cryosphere*, 8, 1289-1296, doi:10.5194/tc-8-1289-2014.
Eisenman, I., W.N. Meier and J.R. Norris. 2014b. Supplement of: A spurious jump in the satellite record: has Antarctic sea ice expansion been overestimated? *The Cryosphere*, 8, 1289-1296, doi:10.5194/tc-8-1289-2014-supplement.

We apologize for the brevity of the description of the exploratory phase of the analysis. We elaborate on the process here and have added this to the text. There are several different ways to test for structural changes in a time series. We used several and checked using an additional one on the basis of this review. As noted in the text, in an exploratory mode, we split the record into sub-periods and tested the differences in the means. In addition, we examined the data by looking for the years that displayed the largest change in one moving three year average to the next. The two largest changes that were already not included in a previously identified six year period (two three-year moving average periods) using this measure were 1988-1989 and 2006-2007. The next stand-alone change had a much smaller change and the magnitude of the change was relatively similar to the next few candidates. While there is no particular justification for choosing the top 2, 3 or N values, the first two stood out as noticeably different in magnitude than the ones following and suggested reasonably long periods. We also ran a statistical test of the quality of various models of the time series. For example, we tested two structural shifts using a model that compared the statistical significance of a model that defined three different means and no trend to a model that defined a single trend. The former model was significant and the trend was not.

In order to increase confidence in the signal, we have now applied the method of Rodionov et al. (2004) to the NASA Team (NSDIC) record, which yielded the same two breakpoints. While it also suggested a third breakpoint around 2001 (which is

also suggested in the graph in Figure 3), using that breakpoint would create a rather small structural break. In addition, a breakpoint at around 2001 was not as strongly supported by the three year rolling average test. We therefore limited our analysis to two breakpoints. We have also added this new analysis to the paper.

Rodionov, S.N. 2004. A sequential algorithm for testing climate regime shifts. Geophysical Research Letters, 31, L09204, doi:10.1029/2004GL019448.

As noted in the manuscript, however, the short record of the satellite derived time series is a limitation. In addition, as raised by Eisenman et al. (2014) and the reviewer, there is potential for processing artifacts. As noted on page 7, line 31, we do not rule out the possibility that these shifts may be associated with artifacts in the satellite record. As the reviewer observes, it is difficult to obtain a truly independent record of the Arctic sea ice record, which is why we often use the term "quasi-independent". That said, the Barrow-Prudhoe Bay navigable days time series is a good candidate for independence, and yielded the same behavior. Nevertheless, we agree that it is not possible to completely rule out the conclusion that these shifts are *not* driven by physical processes. We feel that we have taken this as far as we can in the data realm and are conducting ongoing analysis using a model.

Finally, if as we suggest the structural shifts *are* physical in nature, there is no reason to expect that they should be periodic. If these shifts represent, for example, tipping points in ice thickness, then there is no inherent oscillation in the system that is likely to generate this behavior (although of course that is an hypothesis that could be tested in a climate model.)

Minor revisions

2, 24-25: Note that the CDR simply combines the NASA Team and Bootstrap estimates, so the different products are not completely independent.

> The reviewer is correct in that there is a merged product 1979-2014 and a CDR 1988-2014. It may be semantics, these products in our view are less of a merging of Team and Bootstrap and more of choosing the least bad one, at least in September. But indeed, none of these products are independent and we have added a clarification to that effect.

2, 27: The "merged" product uses data that underwent manual quality control over its entire record, 1978-2014, not just the period prior to 1988.

> We have corrected this error.

2, 30-31: It is true that PM products do not detect ice well at low concentrations, but the 15% concentration threshold is also selected based on the relatively low spatial resolution of the sensors, which makes any ice edge somewhat "fuzzy". I believe the 15% threshold was originally defined because it was found to best match the "true" ice edge during validation studies.

We have added this criterion.

3, 10: I think HadISST2 uses OSI-SAF concentrations since 1979, at least according to Titchner et al., 2014.

Titchner and Rayner (2014) confirm that HadISST2 uses OSI-SAF (Ocean and Sea Ice Satellite Applications Facility) passive microwave retrievals, which is product using SMMR, SSM/I, SSMIS, and EMSR (1972-1979). HadISST2 does *not* use EMSR. Interestingly, the preferred method for processing the data is a combination of the NASA Bootstrap method and the Bristol algorithm (Smith 1996, Hanna and Bamber 2001). The Bristol algorithm applies to Antarctic sea ice, meaning the NH product is effectively Bootstrap.

Hanna, E., and J. Bamber (2001), Derivation and optimization of a new Antarctic sea-ice record, Int. J. Remote Sens., 22(1), 113–139, doi:10.1080/014311601750038884.

Smith, D. M. (1996), Extraction of winter total sea ice concentration in the Greenland and Barents Seas from SSM/I data, Int. J. Remote Sens., 17, 2625–2646.

Titchner, H. A., and N. A. Rayner (2014), The Met Office Hadley Centre sea ice and sea surface temperature data set, version 2: 1. Sea ice concentrations, J. Geophys. Res. Atmos., 119, 2864–2889, doi:10.1002/2013JD020316.

3, 13: It's worth noting that Radarsat data started in 1995. This represents a substantial change in NIC's chart. Before then, they used a lot of PM data; after, it was much less. Also it would be good to include a journal reference for the NIC charts, e.g., Dedrick et al.

We have added this observation and included the reference.

4, 4: Not sure what you mean by "rasterized"? The source data are already on a grid. Maybe rephrase this sentence.

The original ice data is on a grid but then the tracking scheme is Lagrangian and hence no longer grid based. As a result, the derived ice age must then be re-gridded.

5, 23: The HadISST does use NIC charts, but only as a bias correction.

Titchner and Rayner (2014, see reference above) on page 2869 state that "The NIC chart record appears stable relative to OSI SAF after this time; therefore, we chose to use the NIC chart data from 1995 onward as the representation of the "true state" against which to adjust the relative biases in the other data sources used." For this reason, we feel that the characterization "only as bias correction" underplays the role of the NIC charts. We clarified this role on the manuscript.

5, 25: I think the resolution likely accounts for a lot of the differences between the 100 km HadISST and the 25 km PM records.

We agree, and have added a comment to this effect.

7, 15: It would seem more relevant to use the same period as the PM record, i.e., 1979-1988 instead of 1953-1988, etc. Would that potentially affect the analysis?

This would not affect the analysis except near the start and end of the records, which is why we note that significance is lower in the case of the passive microwave data records for the late 1980's breakpoint. But the breakpoint is still detected, using the multiple means described above.

7, 25: Adding the Eisenman reference (above) somewhere here would be useful

The Eisenman et al. (2014) references have been added to the discussion in several locations.

7, 26: What is meant by "errors in the data stream in the first months of the SSMI instrument"? Do you mean the first SSMI, launched in 1987, on DMSP F8? I'm not aware of such errors. The Harvey reference mentions recent errors in 2016, but these don't seem relevant to analysis at hand.

Yes, this is correct (the first SSM/I, launched in 1987, on DMSP F8). We have this information in an email from J. Stroeve on May $2^{nd}$ 2016, in which she documented the small error, and she estimated it would not impact the results by more than 15% (and hence unlikely to be the cause the structural shifts we detected.) We have removed the other (Harvey) reference, as the reviewer is correct – it is discussing a more general issue.

7, 33: The authors say Week 37 is used for ice age. Is this consistent past the sea ice "birthday"

According to the documentation for the sea ice age product, the sea ice age "birthday" is defined by the September minimums; that is, it does not increment in age until a minimum is passed. This typically occurs in September, according to the documentation, although not, as the reviewer notes, on a particular week. Our figure 5 is based on the data in the week 37 file; we have not applied interpolation or extrapolation in order to maintain consistency with the manner in which we have calculated average ice areas.

8, 4: The age algorithm doesn't so much "favor" the oldest ice as it effectively records only the oldest ice in a given cell.

We have changed this phrasing.

Figure 2: The caption says the units are % open water, but the y-axis is 0-0.8. I assume the y-axis is actually sea ice fraction (unit-less, 0-1 range)?

We have corrected the caption.

Figure 3: Same comment as for Figure 2. Both figures would also benefit from a bigger font for the axis numbers and legend.

The caption does use fraction, not %. We will supply larger fonts in the publication quality figures.

References: Just a style comment and this may be a formatting standards issue, but it would be helpful to indent the Reference list. It's very hard to pick out individual references.

We have conformed the Cryosphere format template which did not include indentation of references.

---

## Referee Comment (RC2) · Anonymous Referee #2 · 22 Aug 2016

August 22, 2016

I see on the discussion site that an anonymous review for this paper was submitted on June 21, and the authors submitted a response on July 8. I have NOT read either the review or the response, and therefore the following review is not influenced by either one.

This work uses monthly-averaged passive microwave sea-ice concentration data to examine the area of open water in two broad regions of the Arctic – the Pacific sector and the Atlantic sector – for the period 1979-2014. The researchers find that the time series of open water area in September in the Pacific sector undergoes a significant shift in its mean value in 1988 and again in 2007. The Atlantic sector shows shifts in the same years, though weaker. The researchers call into question the idea of fitting linear trend lines to time series of sea ice or open water.

[Figure]

I share the researchers' contention that linear trend lines are not necessarily the best way to fit time series of sea ice or open water, and in fact I have looked at alternative curve-fitting options myself, including breakpoints, so I am sympathetic to this basic point.

Unfortunately this paper suffers from multiple fatal shortcomings. We are not told how the breakpoint years (1988 and 2007) are identified in the data. The descriptions of the regression models are impossible to follow, and no equations are given. The analysis appears to compare regression models with different numbers of free parameters, and therefore it's not clear whether the better fit is simply due to more degrees of freedom. The work analyzes open water area, which is just the additive inverse of sea-ice area, which has been extensively studied using the same data in the same geographical locations – so why don't we see breakpoints in the time series of sea-ice area? (e.g. see the figures in Parkinson and Cavalieri, J. Geophys. Res., 113, doi:10.1029/2007JC004558, 2008); some discussion is warranted. The self-organizing maps and their application are not well explained. There are many odd sentences in the paper.

Main Comments

We are not told how the breakpoint years are identified. Page 5 line 32 says, "There is an ostensible breakpoint at 1988..." and page 6 line 2 says, "A second breakpoint can be identified in 2007..." How? This is a key part of the analysis, but we are left completely in the dark.

The description of the regression model (page 6 line 5) says, "A model comprised of a constant, a trend variable, and a variable with the three means for each period..." What does this mean? A simple equation would probably clarify everything, but there are none. How many free parameters does this model have?

Page 4, lines 22-30. This is the paragraph about the self-organizing map analysis of sea level pressure. Lines 23-24: "daily sea level pressure anomalies (that is, summer monthly differences...)" – this is really confusing. Also, why are the maps arranged in

a 5 x 4 array, as opposed to (say) a 4 x 3 array or a 6 x 5 array? What do the rows and columns of the array represent? Page 8 line 21, how is Node (3,1) constructed, for example?

Page 6, lines 5-10. Here different regression models are compared, with one having a higher "adjusted $R^2$" than the other, but it's not clear whether the model with the better fit simply has more free parameters. Page 6 lines 12-34, impossible to figure out what's going on without equations to help.

Minor Comments (in page order)

Page 1, lines 23-24. This is an extremely strange sentence, the meaning of which is unclear.

Page 1, lines 30-31. Krupnik & Jolly, AMAP, and Liu & Kronback are not listed in the References. Page 5 line 14, Meier 2005 is not listed in the References.

Page 2, line 15. What does "artifacts" refer to here?

Page 3, lines 23-25. A map of the regions would be helpful.

Page 4, line 3. IABP needs a reference.

Page 5, line 2. "The open water fraction area..." Which one, fraction or area?

Page 5, line 30. I know that $R^2$ is the squared correlation of the fit, but what is the "adjusted" $R^2$? Page 7 line 16: "had a negative adjusted $R^2$" – strange that squared correlation can be negative; what sort of adjustment is done to $R^2$, and why?

Page 7, line 2. What does it mean for a time series to be "temporally uniform"?

Page 7, lines 26-27. Concerning possible errors in the SSM/I data, the authors cite personal communication and an article in the Washington Post. Aren't the errors actually documented somewhere?

Page 8, lines 1-2. Regarding the ice age anomalies in the first 6 years of the record,

"this is likely an artifact of the data product's initialization" – isn't this documented somewhere?

Page 8, line 33. "A long-term, high quality and temporally consistent record of Arctic open water remains an elusive goal." This is a strange sentence that needs further discussion.

Page 9, lines 9-10. "the processes associated with these shifts can only be tested independently using a modeling approach" – hasn't this (modeling approach) been done before?

Figure 1. The scale bar is too small to read.

Figure 2. The caption says "open water (%)" but the vertical scale runs from 0 to 0.8, suggesting that it is fraction rather than percent. The legends are too small to read. Why is some data plotted as lines and other data plotted as points? The colors are difficult to distinguish.

Figure 3. The axis labels are too small to read easily. The vertical axis is labelled "Open Water Area [%]" but the vertical scale runs from 0 to 0.6, suggesting that it is fraction rather than percent.

Figure 5. I can't figure out what the message is here. What is the reader supposed to notice? The colors are difficult to distinguish.

Conclusion

It is impossible to follow the analysis in this work, and I have low confidence that it is done correctly. There should be some discussion of why breakpoints are found for open water area but other researchers have not found breakpoints for sea-ice area. Parts of the Introduction and Discussion appear to be irrelevant, speculative, or just plain strange. I must emphatically recommend that this paper be rejected.

---

## Author Comment (AC2) · 24 Aug 2016

A full response to major and minor comments by the reviewer is attached in a PDF.

What follows is a response to the major comments by Reviewer 2.

We appreciate the comments by Anonymous Referee #2, who has highlighted important areas with additional detail would clarify the work. Since we acknowledge Anonymous Referee #2 has not read the review by and response to Anonymous Referee #1, and hence there is some repetition in our response here to our previously posted response, due to similar recommendations being made. That said, Anonymous Referee #2 has highlighted important areas with additional detail would clarify the work. In the response below, we address each of the main comments by Anonymous Referee #2, and where noted added additional detail to the revised paper.

Main Comments

1. As noted in the response to Anonymous Referee #1, we apologize for the brevity of the description of the exploratory phase of the analysis. We have documented in the revised text the three methods that were used to identify statistically significant breakpoints in the timeseries described in section 2, as follows:

\* We split the record into sub-periods of three years' duration and ranked the differences in the moving averages.

\* Using the top ranked differences for non-overlapping time periods, we tested a series of models for significance. The equations are provided in the attachment.

\* In our revision, we implemented the breakpoint detection approach suggested by Rodionov (2004) as an additional check on our results.

As suggested by the referee, we have added these equations and a more detailed description of the analysis process to the Method description, and we agree that this makes the results much easier to follow. As is evident, it was a nested series of decisions that was required to fully convince us fully that a shift in the mean was the appropriate model for the open water time series over a simple linear trend.

2. The self-organizing map is an unsupervised classification technique based on neural networks. There is an extensive literature on this approach, which does not vary appreciably from application to application. Since this classification approach is analyzed in great detail in Lynch et al. [2016] and is not the main focus of the paper, and is analyzed in great detail in Lynch et al. [2016] and hence we chose not to describe the process in great detail in this publication. However, Lynch et al. [2016] was not included in our original reference list. We apologize for this omission and have corrected it in the revised manuscript. In addition, we have provided some more detail regarding the technique, as follows:

"The allocation is achieved by minimizing the Euclidean distance between a vector representing the sea level pressure matrix and the vector representing that node. After each allocation, the entire array of nodes is adjusted to maximize the Euclidean distance between nodes. A 5x4 array was selected to provide a balance between a practical minimum of nodes and a desirable maximum of variability represented, as described in Lynch et al. [2016]."

Further, we have tried to clarify the clause that the referee found confusing as follows: "In this analysis, summer daily sea level pressure anomalies (that is, the differences between a July, August or September day and the entire period average for July, August and September). . ."

We have now also added the following to the reference list: Lynch, A.H., M.C. Serreze, E.N. Cassano, A.D. Crawford and J. Stroeve, 2016: Linkages between Arctic summer circulation regimes and regional sea ice anomalies. J. Geophys. Res. 121 (13), 7868-7880. DOI: 10.1002/2016JD025164.

3. Because the third regression has two degrees of freedom, the referee is correct in noting that using $r^2$ alone would result in the selection of the model that has the most free parameters. Hence the use of an adjusted correlation coefficient was required to account for the increase in explanatory variables. Since adjusted $r^2$ adjusts for the number of predictors in the model, it is smaller than $r^2$ alone and can become negative. While $r^2$ is a measure of fit, the adjusted $r^2$ is a measure of the suitability of alternative models.

Please also note the supplement to this comment:
http://www.the-cryosphere-discuss.net/tc-2016-108/tc-2016-108-AC2-supplement.pdf

**Supplement:**

Response to: Interactive comment by Anonymous Referee #2 on The Cryosphere Discuss., doi:10.5194/tc-2016-108, 2016.

We appreciate the comments by Anonymous Referee #2, who has highlighted important areas with additional detail would clarify the work. Since we acknowledge Anonymous Referee #2 has not read the review by and response to Anonymous Referee #1, and hence there is some repetition in our response here to our previously posted response, due to similar recommendations being made. That said, Anonymous Referee #2 has highlighted important areas with additional detail would clarify the work. In the response below, we address each of the main comments by Anonymous Referee #2, and where noted added additional detail to the revised paper.

Main Comments

1. As noted in the response to Anonymous Referee #1, we apologize for the brevity of the description of the exploratory phase of the analysis. We have documented in the revised text the three methods that were used to identify statistically significant breakpoints in the timeseries described in section 2, as follows:
- We split the record into sub-periods of three years' duration and ranked the differences in the moving averages.
- Using the top ranked differences for non-overlapping time periods, we tested a series of models for significance as follows:

$$y = \alpha + \beta T + \varepsilon \qquad (1)$$
$$y = \alpha + \beta \pi_s + \epsilon \qquad (2)$$
$$y = \alpha + \beta_1 T + \beta_2 \pi_s + \varepsilon \qquad (3)$$

    where $\alpha$ is a constant, $T$ is the linear trend, and $\pi$ is the sub-period mean. The free parameters are the $\beta$, and $\varepsilon$ is minimized.
- In our revision, we implemented the breakpoint detection approach suggested by Rodionov (2004) as an additional check on our results.

As suggested by the referee, we have added these equations and a more detailed description of the analysis process to the Method description, and we agree that this makes the results much easier to follow. As is evident, it was a nested series of decisions that was required to fully convince us fully that a shift in the mean was the appropriate model for the open water time series over a simple linear trend.

2. The self-organizing map is an unsupervised classification technique based on neural networks. There is an extensive literature on this approach, which does not vary appreciably from application to application. Since this classification approach is analyzed in great detail in Lynch et al. [2016] and is not the main focus of the paper, and is analyzed in great detail in Lynch et al. [2016] and hence we chose not to describe the process in great detail in this publication. However, Lynch et al. [2016] was not included in our original reference list. We apologize for this omission and have corrected it in the revised manuscript. In addition, we have provided some more detail regarding the technique, as follows:

"The allocation is achieved by minimizing the Euclidean distance between a vector representing the sea level pressure matrix and the vector representing that node. After each allocation, the entire array of nodes is adjusted to maximize the

Euclidean distance between nodes. A 5x4 array was selected to provide a balance between a practical minimum of nodes and a desirable maximum of variability represented, as described in Lynch et al. [2016]."

Further, we have tried to clarify the clause that the referee found confusing as follows:
"In this analysis, summer daily sea level pressure anomalies (that is, the differences between a July, August or September day and the entire period average for July, August and September)…"

We have now also added the following to the reference list:
Lynch, A.H., M.C. Serreze, E.N. Cassano, A.D. Crawford and J. Stroeve, 2016: Linkages between Arctic summer circulation regimes and regional sea ice anomalies. *J. Geophys. Res.* 121 (13), 7868-7880. DOI: 10.1002/2016JD025164.

3. Because the third regression has two degrees of freedom, the referee is correct in noting that using $r^2$ alone would result in the selection of the model that has the most free parameters. Hence the use of an adjusted correlation coefficient was required to account for the increase in explanatory variables. Since adjusted $r^2$ adjusts for the number of predictors in the model, it is smaller than $r^2$ alone and can become negative. While $r^2$ is a measure of fit, the adjusted $r^2$ is a measure of the suitability of alternative models.

Minor Comments

Page 1, lines 23-24. This is an extremely strange sentence, the meaning of which is unclear.

We appreciate this suggestion for clarity and have rewritten the sentence and provided an appropriate citation.

Page 1, lines 30-31. Krupnik & Jolly, AMAP, and Liu & Kronback are not listed in the References. Page 5 line 14, Meier 2005 is not listed in the References.

We apologize for the missing references. All have been added to the reference list.

Page 2, line 15. What does "artifacts" refer to here?

While in common usage the word "artifact" tends to mean a human-made object, in this case, we are using the term "artifact" using the second meaning of the word, i.e., detection of a spurious signal arising from extrinsic characteristics of the record, such as the length of a record or the dates on which is starts or ends. (ref, Merriam-Webster dictionary, definition 2.)

Page 3, lines 23-25. A map of the regions would be helpful.

Since the regions are simply longitudinally bound sectors we did not include a map for space reasons, but we have created one (shown below), and will include it in the final revision if the editor requests it. The regional definitions follow the conventions defined by *Gloersen et al.* [1992] and are in use by operational centers as formal analysis areas. We have added this citation:

Gloersen, P., W. J. Campbell, D. J. Cavalieri, J. C. Comiso, C. L. Parkinson, and H. J. Zwally, (1992). Arctic and Antarctic sea ice, 1978-1987: Satellite passive-microwave observations and analysis, NASA Spec. Publ., SP-511.

[Figure]

Page 4, line 3. IABP needs a reference.

Thank you for that point of clarification which we missed. We have preplaced the undefined acronym for the International Drifting Buoy Program with the reference:
Pfirman, S.L., R. Colony, D. Nürnberg, H. Eicken, I. Rigor (1997), Reconstructing the origin and trajectory of drifting Arctic sea ice. J. Geophys. Res. Ocean., 102, 12575–12586

Page 5, line 2. "The open water fraction area..." Which one, fraction or area?

We apologize for the error, the correct phrase is "open water area".

Page 5, line 30. I know that Rˆ2 is the squared correlation of the fit, but what is the "adjusted" Rˆ2? Page 7 line 16: "had a negative adjusted Rˆ2" – strange that squared correlation can be negative; what sort of adjustment is done to Rˆ2, and why?

The adjusted $R^2$ formulation is commonly used when considering alternative model specifications as in this paper. The specific formulation of adjusted $R^2$ and the associated discussion of its use can be found in standard statistics or

econometric textbooks. It is always lower than $R^2$, and decreases when a predictor improves a model by less than expected by chance. It can be negative.

Page 7, line 2. What does it mean for a time series to be "temporally uniform"?

A time series is temporally uniform when the procedure for creating the time series is constant over time, as for, for example, a re-analysis. In the text, we have clarified our overly efficient terminology.

Page 7, lines 26-27. Concerning possible errors in the SSM/I data, the authors cite personal communication and an article in the Washington Post. Aren't the errors actually documented somewhere?

We have removed the Washington Post article at the request of Anonymous Referee #1. The errors were discovered in the course of the research conducted in this paper. National Snow and Ice Data Center released a corrected version on July $6^{th}$, after the submission of this manuscript, but we felt it was important to document this problem as it affects the literature that uses the existing data set.

Page 8, lines 1-2. Regarding the ice age anomalies in the first 6 years of the record, "this is likely an artifact of the data product's initialization" – isn't this documented somewhere?

This is documented in the Data section, which describes how the ice age product is initialized (and hence why it may be problematic), and provides the reference for this product (Tschudi et al. 2016).

Page 8, line 33. "A long-term, high quality and temporally consistent record of Arctic open water remains an elusive goal." This is a strange sentence that needs further discussion.

This manuscript and Lynch et al. (2016) describes the ways in which we do not possess a long-term, high-quality and temporally consistent record of ice and open water in the Arctic. We are not sure why this sentence would be considered strange considering the foregoing, but have provided some references at the end of the sentence in the hope that this addresses the referee's concern (Lynch et al. 2016 and Eisenman et al. 2014b).

Page 9, lines 9-10. "the processes associated with these shifts can only be tested independently using a modeling approach" – hasn't this (modeling approach) been done before?

These shifts have not been documented before. The closest effort identified is the Holland et al. (2006) paper, but the time scale of these shifts is much longer than being discussed here (decadal rather than annual). But we are presently starting this modeling work in collaboration with NCAR CSEM researchers.

Figure 1. The scale bar is too small to read.

This has been corrected in the high resolution versions of the figures.

Figure 2. The caption says "open water (%)" but the vertical scale runs from 0 to 0.8, suggesting that it is fraction rather than percent. The legends are too small to read. Why is some data plotted as lines and other data plotted as points? The colors are difficult to distinguish.

The caption been corrected. Some data are plotted as points because they are very similar to the other versions of the record (but not identical). Hence we have included these time series to highlight the differences but as points for clarity.

Figure 3. The axis labels are too small to read easily. The vertical axis is labelled "Open Water Area [%]" but the vertical scale runs from 0 to 0.6, suggesting that it is fraction rather than percent.

This has been corrected.

Figure 5. I can't figure out what the message is here. What is the reader supposed to notice? The colors are difficult to distinguish.

Figure 5 demonstrates the correlation between open water and ice age, as a part of the discussion of likely mechanism for these abrupt shits, a mechanism associated with a threshold in ice thickness, for which ice age is an imperfect proxy. The colors were selected to avoid problems for the red-green colorblind reader and are the same as Figure 3.